# Understanding summertime $H_2O_2$ chemistry in North China Plain through observations and modelling studies

Can Ye[1], Pengfei Liu[2*], Chaoyang Xue[3*], Chenglong Zhang[2], Zhuobiao Ma[2], Chengtang Liu[2], Junfeng Liu[2], Keding Lu[4], Yujing Mu[2*], Yuanhang Zhang[4]

[1] School of Environmental Science and Engineering, Tiangong University, Tianjin 300387, China
[2] Research Center for Eco-Environmental Sciences, Chinese Academy of Sciences, Beijing 100085, China
[3] Max Planck Institute for Chemistry, Mainz 55128, Germany
[4] State Key Joint Laboratory of Environment Simulation and Pollution Control, College of Environmental Sciences and Engineering, Peking University, Beijing, 100871, China

*Correspondence to*: Pengfei Liu (pfliu@rcees.ac.cn) , Chaoyang Xue (ch.xue@mpic.de), Yujing Mu (yjmu@rcees.ac.cn)

**Abstract.**

Hydrogen peroxide ($H_2O_2$) is a key atmospheric oxidant, crucial for oxidation capacity and sulfate production. However, its chemistry remains understudied compared to ozone ($O_3$), limiting our understanding of photochemical pollution. In summer 2016, atmospheric peroxides and trace gases were measured at a rural site in the North China Plain. $H_2O_2$ was the dominant peroxide ($0.62\pm0.80$ ppb), constituting 69% of total peroxides. It exhibited diurnal variation similar to peroxyacetyl nitrate (PAN) and $O_3$, indicating photochemical production. The $O_3/H_2O_2$ ratio was higher on high-particle days, suggesting $H_2O_2$ uptake by particles reduces its concentration. A box model with default gas-phase chemistry overestimated $H_2O_2$ by a factor of 2.7, and including particle uptake of $H_2O_2$ (uptake coefficient: $6\times10^{-4}$) improved agreement with observations, although we note this value carries some uncertainty related to the assumed $HO_2$ uptake coefficient.

$HO_2$ recombination contributed 91% of $H_2O_2$ production, with a peak rate of 1 ppb h$^{-1}$. Major removal pathways included particle uptake (69%), dry deposition (25%), OH reaction (4%), and photolysis (2%). Relative incremental reactivity (RIR) analysis showed that reducing $NO_x$, $PM_{2.5}$, and alkanes increased $H_2O_2$, while reducing alkenes, aromatics, CO, and HONO decreased it, with alkenes having the strongest effect. $H_2O_2/NO_z$ ratios (>0.15 in 82% of cases) indicated $O_3$ formation was in a transition and $NO_x$-sensitive regime, emphasizing the need for VOC and further $NO_x$ reductions to mitigate both $H_2O_2$ and $O_3$ pollution. These findings improve our understanding of $H_2O_2$ chemistry and provide insights for mitigating photochemical pollution in rural North China.

## 1 Introduction

The atmospheric oxidation capacity is a critical determinant of atmospheric self-cleaning, influencing the residence time and persistence of pollutant gases. Quantifying this capacity is essential for elucidating the lifetimes of pollutants, the formation of aerosols, and their subsequent radiative forcing effects. Hydrogen peroxide ($H_2O_2$) serves as a significant atmospheric

oxidant, primarily generated through the recombination of hydroperoxyl radicals ($HO_2$), which are themselves derived from reactions involving hydroxyl radicals (OH), volatile organic compounds (VOCs), and carbon monoxide (CO). Consequently, the formation of $H_2O_2$ is intrinsically linked to atmospheric oxidation capacity, with its concentration serving as a direct indicator of the intensity of this capacity. Furthermore, as $H_2O_2$ represents a terminal product in the ozone ($O_3$) formation chain reaction, its concentration can be utilized to assess the sensitivity of $O_3$ production to precursors (Sillman, 1995; Reeves and Penkett, 2003; Nunnermacker et al., 2008; He et al., 2010). Owing to its strong oxidative potential and high Henry's law constant, $H_2O_2$ readily dissolves in cloud droplets, where it oxidizes sulfur dioxide ($SO_2$) to form sulfuric acid ($H_2SO_4$), thereby contributing to sulfate aerosol formation and acid rain deposition (Calvert et al., 1985). Research indicated that $H_2O_2$-mediated oxidation of $SO_2$ in cloud water accounts for 60-80% of global $SO_2$ oxidation (Penkett et al., 1979; Calvert et al., 1985; Sofen et al., 2011). Additionally, recent studies have highlighted the significant role of particle-phase $H_2O_2$ oxidation in sulfate formation during winter (Ye et al., 2018; Ye et al., 2021b; Gao et al., 2024). Given its potent oxidative properties, $H_2O_2$ also poses substantial risks to human health and vegetation (Chen et al., 2010). Thus, a precise understanding of $H_2O_2$ chemistry is imperative for advancing knowledge of atmospheric oxidation processes and for diagnosing underlying secondary pollution formation mechanisms.

Atmospheric $H_2O_2$ concentrations are currently reported to range from 0.1 to 13 ppb (Balasubramanian and Husain, 1997; Walker et al., 2006; Ren et al., 2009; Guo et al., 2014; He et al., 2010; Qin et al., 2018; Fischer et al., 2015; Fischer et al., 2019; Ye et al., 2022; Allen et al., 2022; Zhang et al., 2018), with their spatial and temporal variability governed by a balance between production sources and removal pathways. $H_2O_2$ is generated through both primary and secondary sources. Primary sources of $H_2O_2$ include biomass burning, which can contribute substantially under specific conditions. For instance, Ye et al. (2022) reported elevated $H_2O_2$ concentrations during biomass combustion events, which promote secondary sulfate formation and thereby increase fine particulate matter ($PM_{2.5}$) concentrations. The dominant secondary source is the recombination of $HO_2$ radicals, a process enhanced during summer months due to increased solar radiation, which elevates $HO_2$ concentrations and consequently leads to higher $H_2O_2$ levels. However, under elevated nitrogen oxide (NOx) conditions, nitric oxide (NO) reacts competitively with $HO_2$, suppressing $H_2O_2$ formation and resulting in reduced atmospheric concentrations. Another secondary source involves the ozonolysis of alkenes, which produces Criegee intermediates that can decompose to form $H_2O_2$ (Becker et al., 1990). This pathway is particularly relevant during nighttime and potentially in winter, when photochemical activity is diminished (Lee et al., 2008b). For example, alkene ozonolysis was found to dominate wintertime $H_2O_2$ levels (>70%) (Qin et al., 2018), although the yields are generally low, often below 10%. Additionally, the release of $H_2O_2$ from the particle phase has been proposed as a potential source, though its contribution is considered negligible compared to gas-phase production. Recent studies, however, have highlighted that under polluted conditions, high concentrations of humic-like substances and transition metals can facilitate particle-phase $H_2O_2$ formation, which subsequently partitions into the gas phase, significantly enhancing gas-phase $H_2O_2$ levels (Ye et al., 2021b; Liu et al., 2021).

H$_2$O$_2$ can be removed by photolysis, which not only depletes H$_2$O$_2$ but also serves as a source of hydroperoxyl radicals (HO$_x$). However, due to lower photolysis frequency, the contribution of H$_2$O$_2$ photolysis to atmospheric HO$_x$ production is generally much smaller compared to photolysis of O$_3$, nitrous acid (HONO), and formaldehyde (HCHO). Notably, particle-phase H$_2$O$_2$ photolysis has been identified as a critical source of free radicals within aerosols, accelerating aerosol aging and promoting the formation of secondary pollutants. Rao et al. (2023) further emphasized a significantly accelerated rate for air-water interface H$_2$O$_2$ photolysis, underscoring its importance as a source of particle-phase OH. Dry deposition is another key removal mechanism, leading to a vertical gradient in H$_2$O$_2$ concentrations, with peak levels observed at approximately 2 km above the surface (Watanabe et al., 2016; Klippel et al., 2011). Due to its high solubility, wet deposition through rainwater scavenging also effectively removes H$_2$O$_2$ from the atmosphere. Moreover, laboratory and field studies have demonstrated that heterogeneous uptake by particles can significantly contribute to H$_2$O$_2$ removal under polluted conditions. Qin et al. (2022) reported a maximum uptake coefficient of $2.49 \times 10^{-3}$ for H$_2$O$_2$ by ambient particles, with the uptake coefficient influenced by the concentration of transition metals within the particles.

In addition to H$_2$O$_2$, the atmosphere contains a variety of organic peroxides, such as methyl hydroperoxide (CH$_3$OOH), formed through reactions between HO$_2$ and organic peroxy (RO$_2$) radicals. While H$_2$O$_2$ is the most abundant peroxide in the atmosphere, organic peroxides are recognized as a significant component of secondary organic aerosol (SOA), contributing to aerosol composition and properties. However, due to analytical challenges associated with measuring organic peroxides, most studies on atmospheric peroxides have only focused on H$_2$O$_2$ (Zhang et al., 2012).

Photochemical pollution has emerged as a critical air quality issue in China, impacting both urban and rural regions. H$_2$O$_2$ and O$_3$ are key products of photochemical pollution, and elucidating their chemical behavior is essential for developing effective strategies to mitigate photochemical pollution. However, compared to the extensive research on O$_3$, studies on H$_2$O$_2$ remain limited due to the technical challenges and complexities associated with its measurement. In recent years, O$_3$ concentrations in the North China Plain have exhibited a significant upward trend (Li et al., 2019; Wang et al., 2020; Lu et al., 2020), yet the characteristics of H$_2$O$_2$ in this region remain poorly understood. Furthermore, the implementation of national emission reduction policies has led to a substantial decline in NOx, while VOCs persist at elevated levels (Liu et al., 2023). This shift toward low NOx and high VOCs conditions is more conducive to H$_2$O$_2$ formation. Although photochemical pollution is traditionally considered as an urban phenomenon, recent studies have highlighted its increasing prevalence in rural areas, where pollution levels are gradually approaching those observed in urban areas (Ma et al., 2016). Rural regions typically exhibit lower NOx concentrations than urban areas, creating conditions more favorable for H$_2$O$_2$ production. Despite this, research on H$_2$O$_2$ in rural areas of the heavily polluted North China Plain remains scarce. Consequently, there is an urgent need to investigate H$_2$O$_2$ chemistry in rural environments to inform targeted control strategies for photochemical pollution.

This study is based on a field campaign conducted in a rural area of the North China Plain, during which a comprehensive suite of gaseous (including $H_2O_2$), particulate matter, and meteorological parameters, were measured. Here we investigate the temporal variations of $H_2O_2$, and its relationships with other oxidants (e.g., $O_3$ and peroxyacetyl nitrate, PAN), and preliminarily estimate organic peroxide concentrations. A zero-dimensional box model was employed to examine the influence of particles on the $H_2O_2$ budget and the sensitivity of $H_2O_2$ production to various chemical species. Finally, we explore the potential of $H_2O_2$ as an indicator for determining $O_3$ sensitivity and discuss the control strategy for alleviating photochemical pollution.

## 2 Experiments

### 2.1 Measurement site

The observational experiment was conducted at the Station of Rural Environment, Research Center for Eco-Environmental Sciences (SRE-RCEES, 38°42′N, 115°15′E), located in Dongbaituo Village, Wangdu County, Hebei Province. Situated approximately 180 km southwest of Beijing, the station is surrounded primarily by farmland with no nearby industrial facilities, making it an ideal site for studying typical rural atmospheric conditions. This location has historically served as a key site for numerous large-scale observational campaigns (Tan et al., 2017; Peng et al., 2021). The experiment took place from 6 July 2016 to 12 August 2016, with the primary objective of investigating the underlying causes of photochemical pollution in the rural North China Plain.

### 2.2 $H_2O_2$ measurements

$H_2O_2$ concentrations were measured using the AL-2021 $H_2O_2$ monitor (Aero-Laser) (Lazrus et al., 1986). The instrument operates on the following principle: gas-phase peroxides in ambient air are collected by buffered solution in a glass stripping coil. The trapped peroxides then react with p-hydroxyphenyl acetic acid (POPHA) under the catalysis of peroxidase, producing a fluorescent dimer. This dimer exhibits maximal light absorption at a characteristic wavelength of 320 nm and emits fluorescence with a central wavelength of 400 nm. By continuously monitoring the intensity of this fluorescence signal, the instrument enables online quantitative detection of atmospheric peroxides. To differentiate between $H_2O_2$ and organic peroxides, a dual-channel measurement approach was employed. Channel A measures the total peroxide content, while Channel B incorporates catalase into the absorbent solution to selectively decompose $H_2O_2$, thereby measuring only organic peroxides. The $H_2O_2$ concentration is determined by the difference in signals between the two channels. Although Channel B provides an approximation of organic peroxides, it is important to note that the percentage of organic peroxides reported in this study represents a lower limit, as the collection efficiency of the stripping coil technique varies significantly among different organic peroxide species. While $H_2O_2$ is efficiently trapped due to its high solubility (Henry's law constant: ~$10^5$ M atm$^{-1}$), many organic peroxides such as methyl hydroperoxide (MHP) have substantially lower solubilities (Henry's law

constant: $3 \times 10^2$ M atm$^{-1}$), resulting in lower collection efficiencies. Additionally, the catalase used to differentiate between $H_2O_2$ and organic peroxides may not completely discriminate between certain hydroperoxide species, further contributing to uncertainty in organic peroxide quantification. The detection limit of the $H_2O_2$ measurement instrument is 50 ppt, with an uncertainty of 10%. To ensure the stability of the instrument's operation, regular calibrations are performed at fixed intervals.

In several previous field experiments (Ye et al., 2018; Ye et al., 2021b; Ye et al., 2021a; Liu et al., 2021), this instrument has been successfully utilized to measure atmospheric $H_2O_2$, demonstrating high reliability and consistent operational stability.

## 2.3 Other species

NOx, $O_3$, $SO_2$, $PM_{2.5}$, CO, and total reactive nitrogen (NOy) were measured using commercial instruments from Thermo Electron. Volatile organic compounds (VOCs) were quantified by gas chromatography with a flame ionization detector (GC-

FID), while nitrous acid (HONO) was measured using a long-path absorption photometer (LOPAP) from QUMA. The aerosol surface area density was calculated by combining data from a scanning mobility particle sizer (SMPS) and an aerodynamic particle sizer (APS). PAN was analyzed using gas chromatography with electron capture detection (GC-ECD). Gas-phase meteorological data were collected using a portable meteorological station (Model WXT520, Vaisala, Finland). The photolysis rate constant of $NO_2$ ($j(NO_2)$) was measured directly, and other photolysis rate constants were derived using

the Tropospheric Ultraviolet and Visible (TUV) radiation model, scaled based on $j(NO_2)$ measurements. Detailed information on the experimental instruments is provided in Table S1.

## 2.4 Box model descriptions

A zero-dimensional box model based on the RACM2-LIM1 mechanism was employed to investigate the sources and removal mechanisms of $H_2O_2$. This model is widely recognized for its ability to accurately model HOx radicals (Tan et al.,

2017; Ma et al., 2022). Given that the $HO_2$ is a critical precursor for $H_2O_2$ formation, the model's strong performance in simulating free radicals provides confidence in its ability to reliably simulate $H_2O_2$ concentrations. The model was constrained using input parameters including photolysis rate constants ($j(NO_2)$, $j(O^1D)$, $j(HONO)$, $j(H_2O_2)$, $j(HCHO)$), VOCs, NO, $NO_2$, $O_3$, HONO, methane ($CH_4$), CO, and meteorological data (temperature, relative humidity, and pressure). VOCs were categorized into different reactivity-based groups according to their reaction rates with OH, as detailed in Table S2.

The dry deposition rate constant for $H_2O_2$ was set to $3 \times 10^{-5}$ s$^{-1}$, and boundary layer heights were derived from the hybrid single-particle Lagrangian integrated trajectory (HYSPLIT) model.

The simulation focused on the period from 24 July to 3 August, selected for its stable meteorological conditions, characterized by low wind speeds and predominantly static weather. During this period, the observed trends in $H_2O_2$ concentrations exhibited consistent patterns, suggesting that local photochemical processes were the primary source of $H_2O_2$.

This makes the selected timeframe ideal for exploring $H_2O_2$ sources using the box model. Additionally, elevated $PM_{2.5}$ concentrations during this period provided an opportunity to investigate the potential influence of particle uptake on $H_2O_2$ removal. The rate coefficient of $H_2O_2$ uptake by particles was parameterized as equation 1:

$$k = 0.25 \times c \times \gamma \times S_a \qquad \text{Eq. 1}$$

Here c is mean molecular speed of $H_2O_2$, $\gamma$ is the $H_2O_2$ uptake coefficient, and $S_a$ is aerosol surface area density.

To assess the contributions of different precursors to $H_2O_2$ production, Relative Incremental Reactivity (RIR) analysis was conducted. RIR was calculated using the following equation:

$$\text{RIR}(X) = \frac{\dfrac{\Delta H_2O_2(X)}{H_2O_2}}{\dfrac{\Delta C(X)}{C(X)}} \qquad \text{Eq. 2}$$

In Eq.2, X represents the primary pollutants that may influence $H_2O_2$ concentrations. $H_2O_2$ represents modelled $H_2O_2$ in the base case. $\Delta C(X)/C(X)$ represents the relative change of primary pollutants. $\Delta H_2O_2(X)/H_2O_2$ represents the relative change of modelled $H_2O_2$ concentrations induced by the reduction of X. Considering the variations in simulated radical concentrations and the deviations in the RIR, a 20% reduction scenario was selected for further analysis. This approach allowed for the quantification of the sensitivity of $H_2O_2$ production to variations in precursor concentrations, providing insights into the key drivers of $H_2O_2$ formation in the rural North China Plain.

# 3 Results and discussion

## 3.1 Time series overview

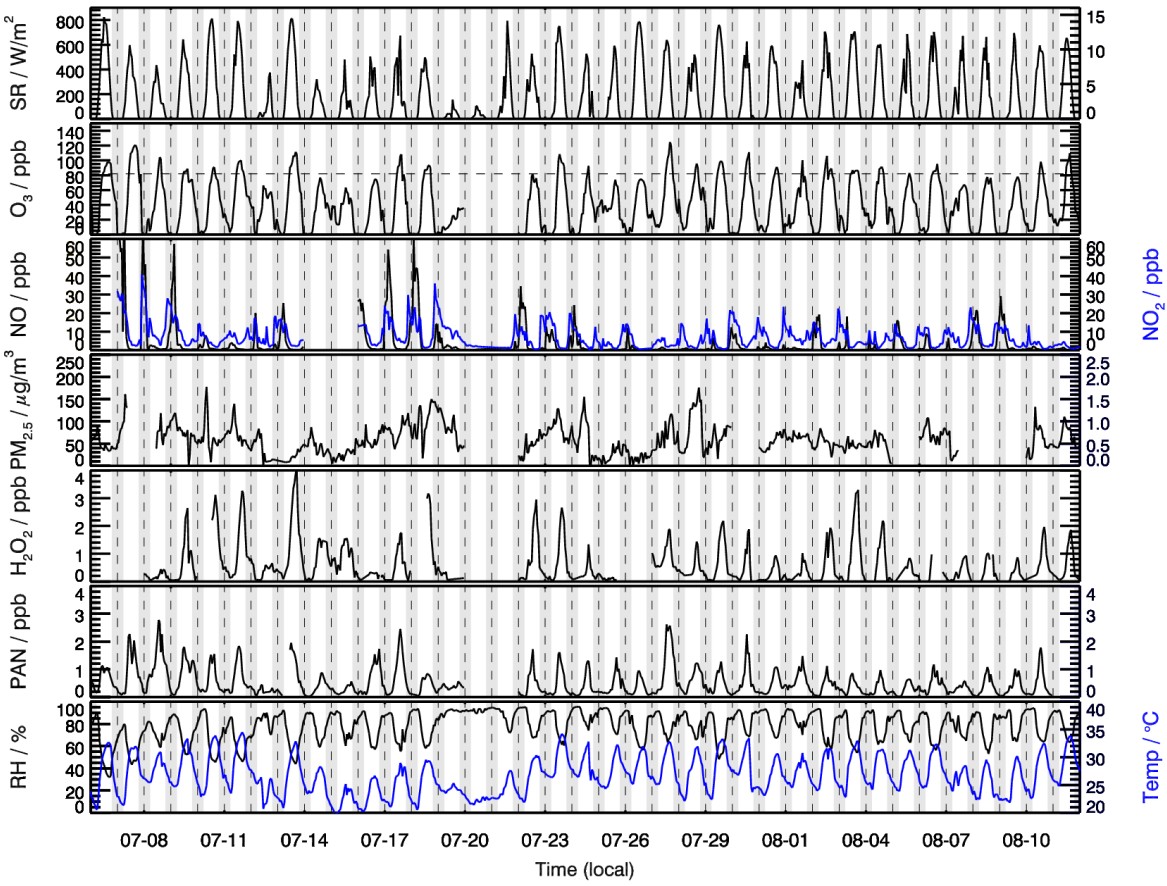

Figure 1. Measurements of $H_2O_2$, other related chemical species and meteorological parameters at SRE-RCEES site during the observation period.

Throughout the observation period, meteorological conditions were characterized by high temperature and relative humidity. High temperature generally increased the rate constants of photochemical reactions, while abundant water vapor enhanced the recombination rate of $HO_2$ and the reaction rate between $O(^1D)$ and water vapor ($H_2O$). The maximum $O_3$ concentration

reached 120 ppb, with the maximum daily 8-hour average (MDA8) frequently exceeding the National Ambient Air Quality Standard (NAAQS) Class-II standard of 82 ppb (25 ℃, 1013 $kPa$). High $O_3$ pollution events often coincided with elevated $H_2O_2$ concentrations (>2 ppb), suggesting that $O_3$ production at this site may be sensitive to NOx. This hypothesis will be further investigated using the $H_2O_2/NOz$ and $O_3/NOz$ in Section 3.6 on $O_3$ sensitivity. NOx concentrations peaked in the morning, driven by factors such as traffic emissions and lower boundary layer height. Daytime NO concentrations were

generally below 1 ppb, while daily peak $H_2O_2$ concentrations exhibited significant day-to-day variability, ranging from

approximately 0.2 ppb to 4 ppb. Higher $H_2O_2$ concentrations were observed during periods of intense solar radiation, indicating that local photochemical reactions play a significant role in $H_2O_2$ production. Notably, elevated $H_2O_2$ levels were only observed when NO concentrations were low, consistent with the known mechanism of $H_2O_2$ formation under low NOx conditions.


The average $H_2O_2$ concentration during the whole observation period was $0.62\pm0.80$ ppb, significantly higher than wintertime concentrations (0.19 ppb) at the same site (Ye et al., 2021b), as summer conditions with high solar radiation intensity and relative humidity are more conducive to $H_2O_2$ production. This average concentration also exceeded summer $H_2O_2$ levels reported in urban areas, such as Beijing (0.27 pb) (Qin et al., 2018) and Hong Kong (0.32 ppb) (Guo et al., 2014),

likely due to lower NOx levels at the rural site, which favor $H_2O_2$ formation. Compared to $H_2O_2$ concentrations reported at rural sites in other countries, the levels observed in this study were lower than that in Kinterbish (Watkins et al., 1995), Whiteface Mountain (1.61 ppb) (Balasubramanian and Husain, 1997). It is worth mentioning that, an average $H_2O_2$ concentration of $0.51\pm0.90$ ppb was reported at the same site in summer 2014 (Wang et al., 2016), lower than the current study's findings, reflecting a potential increasing trend in $H_2O_2$ concentrations over time. In addition, multi-year

measurements at the summit of Mount Tai revealed an increasing trend of $H_2O_2$ concentrations in cloud water from 2014 to 2018 (Li et al., 2020), indirectly indicating rising gas-phase $H_2O_2$ levels in the North China Plain. The significant reduction in NOx emissions in the North China Plain over recent years, while VOC levels remained relatively high or decreased less sharply, has likely shifted the atmospheric chemistry towards conditions more favorable for $HO_2$ recombination, potentially contributing to the observed increasing trend in $H_2O_2$ concentrations. This aligns with the known sensitivity of $H_2O_2$

formation to NOx levels.

Elevated $H_2O_2$ concentrations and high relative humidity in rural areas facilitate the oxidation of $SO_2$ by $H_2O_2$ in both aerosol water and cloud water, contributing to sulfate formation and increased $PM_{2.5}$ levels. During the observation period, the average $PM_{2.5}$ concentration reached 57 μg m$^{-3}$, and the co-occurrence of $PM_{2.5}$ and $O_3$ pollution was frequently observed.

This dual pollution phenomenon suggests that high concentrations of oxidants may play a significant role in driving secondary aerosol formation. PAN, another key secondary oxidant measured in this study, reached a maximum concentration of 2.9 ppb. Similar to $H_2O_2$ and $O_3$, PAN is a product of photochemical pollution, and its temporal trends closely mirrored those of $H_2O_2$ and $O_3$. These trends will be analyzed in detail in the section 3.2. As strong oxidizing agents, $H_2O_2$, $O_3$ and PAN are proven to be damaging to vegetation and human health. Given the high concentrations of these oxidants observed in

this study, photochemical pollution in rural areas poses serious risks to agricultural productivity and human health.

**3.2 Diurnal patterns of three photochemical oxidants**

The average diurnal trends of $H_2O_2$, PAN, and $O_3$ exhibited pronounced daily variations, with concentrations peaking during the daytime and declining at night (Figure 2). These trends closely followed solar radiation patterns, highlighting the

significant contribution of photochemical reactions to their formation. In addition, the pronounced daily variations also
indicated the presence of abundant precursors in the region facilitating the production of $H_2O_2$, PAN, and $O_3$. In the early
morning, as solar radiation intensified, the photolysis of HONO initiated daytime photochemical reactions (R0), generating
peroxyl radicals (R1). These radicals reacted with NO to produce $O_3$ (R2-R5); $HO_2$ recombination underwent bimolecular
recombination to produce $H_2O_2$ (R6); peroxyacetyl radicals (PA) reacted with $NO_2$ to form PAN (R7). These processes led to
a rapid increase in the concentrations of all three oxidants, with peak concentrations reaching 1.8 ppb, 1.2 ppb, and 84 ppb
for $H_2O_2$, PAN, and $O_3$, respectively.

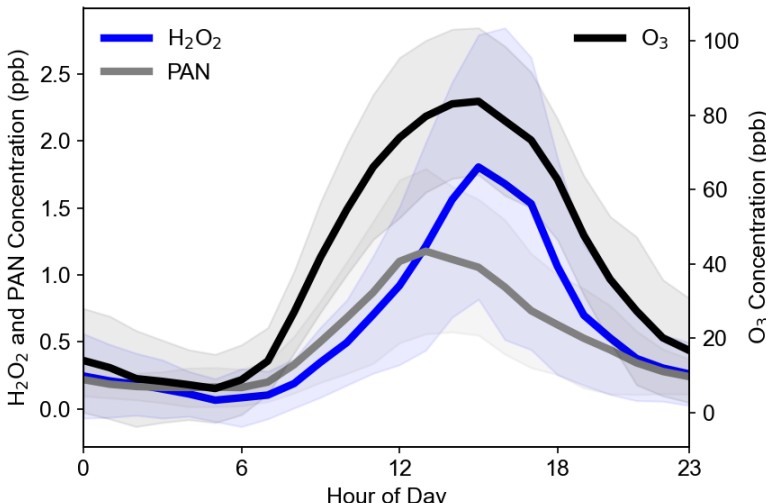

Figure 2. Average diurnal cycles of $H_2O_2$, PAN, and $O_3$ observed throughout the entire campaign period at the SRE-RCEES site.

Despite sharing similar photochemical formation pathways, the peak times of the three oxidants differed due to variations in
their production and removal rates. PAN concentrations peaked around noon, approximately 2–3 hours earlier than $H_2O_2$ and
$O_3$, a phenomenon also observed in previous studies (Lee et al., 2008a). This earlier peak for PAN can be attributed to its
higher thermal decomposition rate at midday. In contrast, the peaks for $H_2O_2$ and $O_3$ both occurred around 16:00. Notably, in
urban areas, $H_2O_2$ peaks often lag behind $O_3$ peaks. For example, observations at the urban Tai'an site in the North China
Plain revealed that $H_2O_2$ peaks occurred approximately 2 hours after $O_3$ peaks (Ye et al., 2021a). This delay can be explained
by $HO_2$ chemistry under varying NOx conditions. Under high NOx condition, $HO_2$ primarily reacts with NO (reaction rate
constant: $8.9\times10^{-12}$ $cm^3$ $molecule^{-1}$ $s^{-1}$ at 298 $K$), whereas under low NOx condition, $HO_2$ undergoes bimolecular
recombination to form $H_2O_2$ (reaction rate constant: $1.5\times10^{-12}$ $cm^3$ $molecule^{-1}$ $s^{-1}$ at 298 $K$). In urban settings, $H_2O_2$ peaks
only occur when NO concentrations drop to around 100 ppt, allowing $HO_2$ recombination to dominate, thus delaying the
$H_2O_2$ peak relative to $O_3$. However, at this rural site, daytime NO concentrations were consistently low, resulting in
simultaneous peaks for $O_3$ and $H_2O_2$.

$$HONO+h\upsilon \rightarrow OH+NO \hspace{4cm} R0$$

$$VOCs+OH \rightarrow RO_2+H_2O \hspace{4cm} R1$$

$$RO_2+NO \rightarrow HO_2+RO_2+OVOC \hspace{3cm} R2$$

$$HO_2+NO \rightarrow NO_2+OH \hspace{4cm} R3$$

$$NO_2+h\upsilon \rightarrow NO+O\left(^3P\right) \hspace{4cm} R4$$

$$O\left(^3P\right)+O_2 \rightarrow O_3 \hspace{4.5cm} R5$$

$$HO_2+HO_2+M \rightarrow H_2O_2+H_2O+M \hspace{2.5cm} R6$$

$$CH_3C(O)OO+NO_2+M \rightarrow CH_3C(O)OONO_2(PAN)+H_2O+M \hspace{1cm} R7$$

Following their peaks, the concentrations of all three oxidants declined rapidly. For $H_2O_2$, this decrease was primarily driven by dry deposition and, in the evening, enhanced uptake by liquid aerosols formed as relative humidity increased. $O_3$ concentrations dropped due to a combination of dry deposition and NO titration, while PAN levels decreased mainly through thermal decomposition. At night, the absence of photochemical reactions caused all three oxidants to maintain low concentrations.

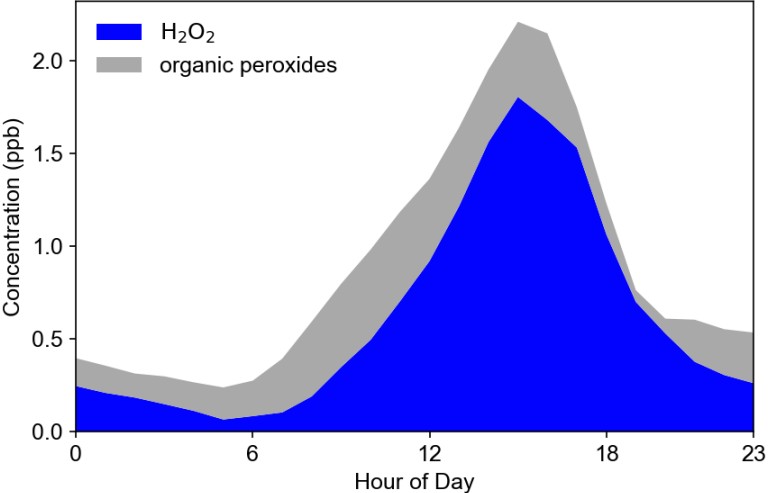

Figure 3. Average diurnal cycles of $H_2O_2$ and organic peroxides (ROOH) observed throughout the entire campaign period at the SRE-RCEES site.

Figure 3 illustrates the average diurnal trends of ROOH and $H_2O_2$. The trends of total peroxides closely align with those of $H_2O_2$, indicating similar production and removal mechanisms. $H_2O_2$ accounts for 69% of the total peroxides on average, while ROOH (0.28 ppb) constitute 31%. This demonstrates that peroxides in rural areas are predominantly dominated by $H_2O_2$, consistent with the findings of Wang et al. (2016) at this site. However, it is important to note that the percentage of ROOH reported in this study represents a lower limit, as not all ROOH are fully captured by the measurement technique. In

contrast, Liang et al. (2013) reported that ROOH accounted for 80% of total peroxides in urban areas such as Beijing. The difference in organic peroxide proportions between Beijing and Wangdu can likely be attributed to variations in chemical conditions, such as differences in VOC compositions, which influence the types and abundances of peroxyl radicals formed.

The diurnal variation in the relative contributions of $H_2O_2$ and ROOH to total peroxides, reflects their distinct production and loss mechanisms. $H_2O_2$ dominates (over 90%) around 19:00 due to strong photochemical production via $HO_2$ recombination during the day, while its contribution drops to ~25% by 05:00 due to nighttime losses (e.g., heterogeneous uptake and dry deposition) without replenishment. In contrast, ROOH contribute more significantly in the early morning, likely due to slower loss rates compared to $H_2O_2$. ROOH such as $CH_3OOH$ (methyl hydroperoxide) have much lower dry deposition

rates—approximately 30 times lower than that of $H_2O_2$-leading to less nighttime loss and a higher relative contribution to total peroxides during early morning hours. The minimum in ROOH concentration observed around 19:00 represents a transitional point. By this time, daytime photochemical production has largely ceased due to diminishing solar radiation, leading to a decline from its afternoon peak as removal processes continue. The subsequent increase in ROOH concentration after 19:00, which makes 19:00 a local minimum, may be attributed to nighttime chemical production primarily through (a)

the ozonolysis of alkenes ($O_3$ + alkenes $\rightarrow \ldots \rightarrow RO_2 \rightarrow ROOH$), and (b) $NO_3$ radical-initiated oxidation of VOCs ($NO_2$ + $O_3 \rightarrow NO_3$; $NO_3$ + VOCs $\rightarrow \ldots \rightarrow RO_2 \rightarrow ROOH$). These processes become major sources of $RO_2$ (and subsequently ROOH) during the night. In contrast, $H_2O_2$ typically continues to decrease throughout the night. Although ozonolysis can also be a source $H_2O_2$, $H_2O_2$ generally has a higher dry deposition velocity than many ROOH species, leading to more efficient net removal overnight. These differences highlight the distinct photochemical dynamics and loss mechanisms of

$H_2O_2$ compared to ROOH, influenced by diurnal variations in radiation, precursor concentrations, and meteorological conditions.

### 3.3 Correlations between different atmospheric oxidants

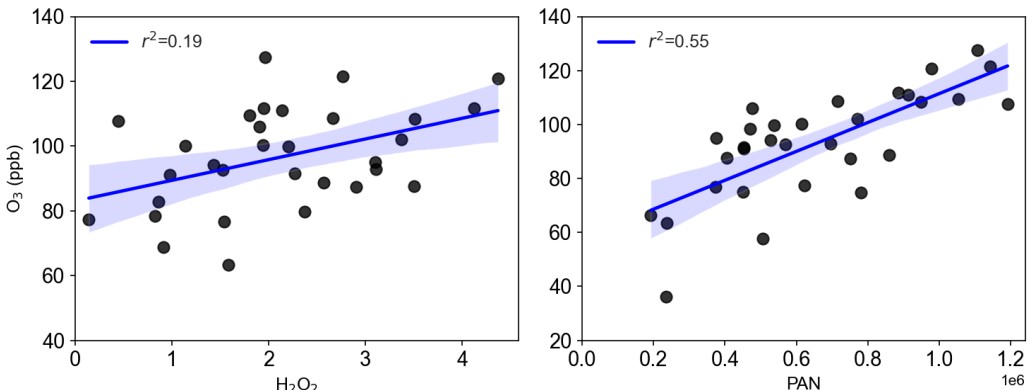

Figure 4. Correlations of $O_3$ daily maximum with $H_2O_2$ and PAN daily maximum


The formation of $H_2O_2$, $O_3$, and PAN is closely linked to VOCs, NOx, and solar radiation. Consequently, their concentrations are typically elevated and well-correlated during photochemical pollution episodes. Here, we investigate the relationships among these oxidants. Figure 4 illustrates the correlations between the daily maximum concentrations of $H_2O_2$, $O_3$, and PAN. A good correlation ($r^2 = 0.55$) was observed between PAN and $O_3$, consistent with previous studies (Lee et al., 2008a; Zhang et al., 2014; Xu et al., 2021; Sun et al., 2020). In contrast, the correlation between $H_2O_2$ and $O_3$ was weak ($r^2 = 0.19$). Prior research has shown positive correlations between $H_2O_2$ and $O_3$ during photochemical pollution due to their shared dependence on VOC and NOx photochemistry (Hua et al., 2008; Takami et al., 2003; Ye et al., 2021a; Guo et al., 2022), while negative correlations have been reported in clean marine boundary layer where $O_3$ photolysis dominates radical production (Ayers et al., 1992). The lack of a positive correlation between $O_3$ and $H_2O_2$ in this rural polluted environment may indicate additional factors influencing $H_2O_2$ concentrations. Notably, heterogeneous uptake by particles has been shown to affect $H_2O_2$ levels (De Reus et al., 2005; Qin et al., 2018), and given the relatively high $PM_{2.5}$ concentrations during the observation period, we hypothesize that heterogeneous loss reduces gas-phase $H_2O_2$, weakening its correlation with $O_3$. Additionally, aqueous-phase reactions in aerosol water or cloud droplets, facilitated by high relative humidity during the campaign, could further reduce gas-phase $H_2O_2$ without affecting $O_3$, contributing to the decoupling of their peak values. While the focus on daytime maxima limits the direct relevance of nighttime chemistry, processes such as alkene ozonolysis or nocturnal deposition could influence background $H_2O_2$ levels, indirectly affecting daytime peaks.

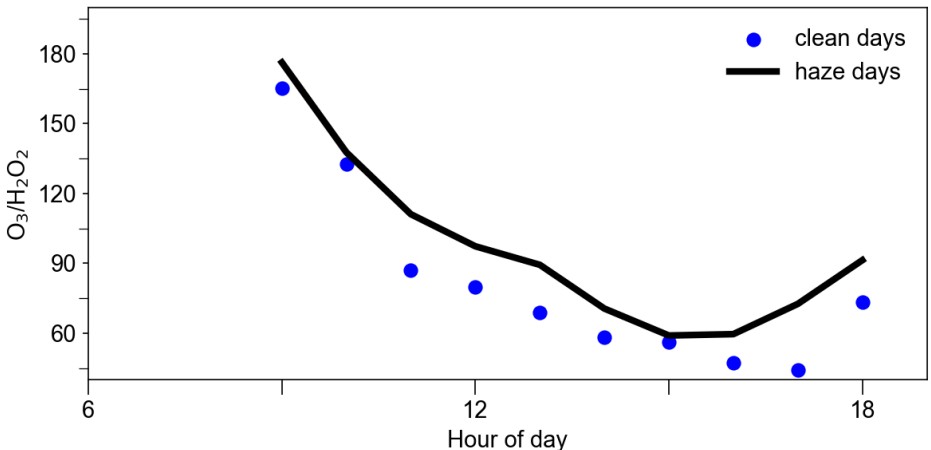

Figure 5. Average $O_3/H_2O_2$ from 9:00 to 18:00 on clean (daily average $PM_{2.5}<50$ μg m$^{-3}$) and polluted days (daily average $PM_{2.5}\geq 50$ μg m$^{-3}$).

To test this hypothesis, we analyzed the $O_3/H_2O_2$ ratio on polluted (daily average $PM_{2.5}<50$ μg m$^{-3}$) and clean days (daily average $PM_{2.5}\geq 50$ μg m$^{-3}$). While $O_3$ and $H_2O_2$ share similar photochemical formation pathways, $O_3$ is less affected by particle uptake. $O_3$ lifetime was estimated to be 13 days with respect to heterogeneous uptake for dust mass concentrations of 1000 μg m$^{-3}$, highlighting the minor role of particle uptake on $O_3$ removal (Tang et al., 2017). If the $O_3/H_2O_2$ ratio remains stable across polluted and clean conditions, heterogeneous uptake likely has minimal impact on $H_2O_2$. However, if the ratio

is higher during polluted periods, it is possible that PM$_{2.5}$ may scavenge H$_2$O$_2$ by heterogeneous uptake. As shown in Figure 5, the O$_3$/H$_2$O$_2$ ratio during peak photochemical hours (9:00–18:00) was markedly higher on polluted days compared to clean days, supporting the hypothesis that heterogeneous uptake by PM$_{2.5}$ significantly reduces H$_2$O$_2$ concentrations. It is important to note that this method provides only a preliminary assessment, as uncertainties exist due to differences in the dependence of H$_2$O$_2$ and O$_3$ on peroxyl radical concentrations and their respective responses to radiation intensity. In

addition, differences in photochemical regimes, potentially driven by varying VOC/NOx ratios between clean and polluted days, could also influence the O$_3$/H$_2$O$_2$ relationship independently of particle uptake effects. In the following section, we further examine the impact of PM$_{2.5}$ on H$_2$O$_2$ budget using a box model.

### 3.4 Investigation on H$_2$O$_2$ budget

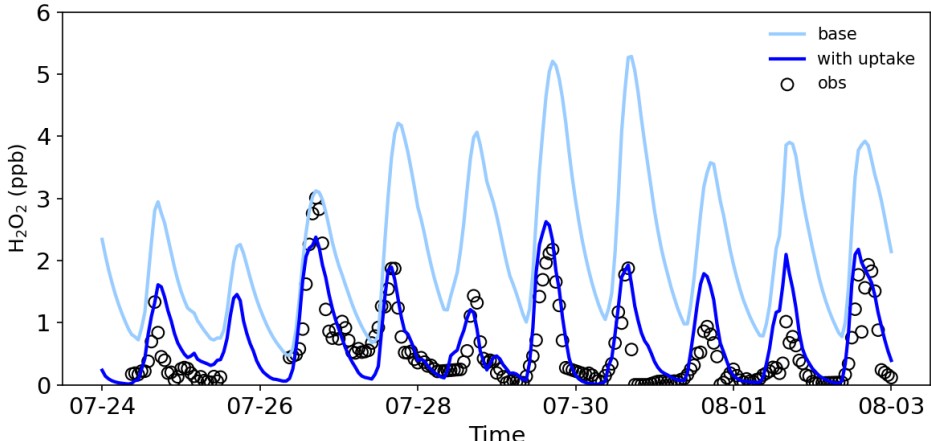

315                       Figure 6. Observed and modelled H$_2$O$_2$ concentrations from 24 July to 3 August.

To better understand the sources and removal mechanisms of H$_2$O$_2$, we employed a box model to simulate its concentrations. As shown in Figure 6, base simulations using the model's default H$_2$O$_2$ source and removal mechanisms overestimated H$_2$O$_2$ concentrations compared to observations, with a simulated-to-measured ratio of 2.7. This discrepancy suggests an

unaccounted removal pathway, consistent with our earlier hypothesis of H$_2$O$_2$ removal by particle uptake. When a parameterized uptake mechanism with an uptake coefficient of $6\times10^{-4}$ was incorporated into the box model, the simulated H$_2$O$_2$ concentrations and trends aligned well with observed values (Fig. 6), confirming the significant role of particle uptake in H$_2$O$_2$ removal in rural areas. This uptake coefficient is comparable with the value ($5\times10^{-4}$) estimated during a dense Saharan dust event (De Reus et al., 2005), and lower than $1\times10^{-3}$ reported by Wang et al. (2016), which may be likely due to

differences in particulate matter composition. Sensitivity tests indicated that an uptake coefficient of $1\times10^{-3}$ resulted in underestimation (Figure.S1), supporting $6\times10^{-4}$ as the optimal value for our study. This coefficient falls within the range ($10^{-4}$-$10^{-3}$) determined in laboratory studies for H$_2$O$_2$ uptake on ambient particles collected on filters or artificial particles

(Pradhan et al., 2010; Romanias et al., 2012; Qin et al., 2022). We believe this value represents a reasonable estimate for the conditions at our sampling site, though we acknowledge that a more dynamic treatment of heterogeneous processes that

accounts for variations in aerosol composition, phase state, and ambient RH would be valuable in future studies.

It should be mentioned that previous studies have demonstrated that considering $HO_2$ by particles can partially explain the discrepancy between observed and modeled $HO_2$ concentrations under low NOx conditions (Kanaya et al., 2007a; Kanaya et al., 2007b; Whalley et al., 2010; Ma et al., 2022), as well as the phenomenon of increasing $O_3$ concentrations with decreasing particulate matter levels (Li et al., 2019). Since $HO_2$ is a precursor to $H_2O_2$, its uptake by particles naturally reduces $H_2O_2$

concentrations. However, laboratory-measured $HO_2$ uptake coefficients exhibit significant variability, ranging from $10^{-5}$ to 0.82, and are strongly influenced by the composition of particulate matter (Thornton et al., 2008; Taketani et al., 2012; George et al., 2013; Lakey et al., 2015). Through analysis of measured radical budget and related parameters, Tan et al. (2020) showed that the $HO_2$ uptake was not important in the North China Plain in 2014, with an uptake coefficient of 0.08. Given that our observational experiments were conducted at the same site with similar particulate matter composition, we

also assumed an $HO_2$ uptake coefficient of 0.08 to investigate its impact on the $H_2O_2$ budget. Under this assumption, we found that an $H_2O_2$ uptake coefficient of $4.5\times10^{-4}$ resulted in a good agreement between modeled and observed $H_2O_2$ concentrations (Figure S1). The results indicate that considering $HO_2$ uptake reduces the $H_2O_2$ uptake coefficient by 25%. Therefore, uncertainties in the $HO_2$ uptake coefficient significantly affect the accurate simulation of $H_2O_2$ concentrations and the estimation of the $H_2O_2$ uptake coefficient. A more precise parameterization scheme for $HO_2$ uptake is critical for models

to accurately assess the global distribution of $H_2O_2$ concentrations and their environmental impacts.

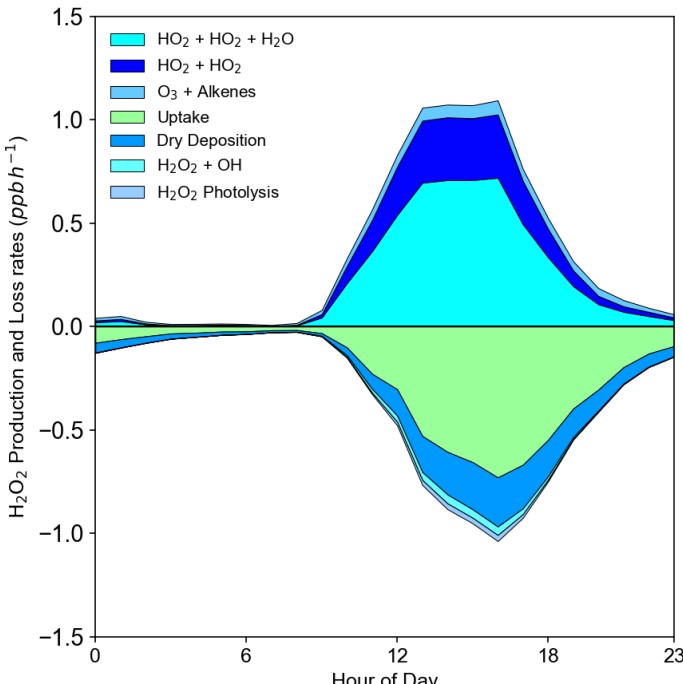

Figure 7. Modelled $H_2O_2$ sources and sinks.

Figure 7 depicts the $H_2O_2$ production rates and removal rates by different pathways. The percentage contribution of different pathways is shown in Figure S2. $HO_2$ bimolecular recombination was identified as the dominant $H_2O_2$ production pathway, contributing to 80% $H_2O_2$ production with a maximum yield of 1.0 ppb h$^{-1}$ at noon. This highlighted rapid photochemical production as the primary driver of $H_2O_2$ pollution in the rural site. In contrast, the reaction of $O_3$ with alkenes accounted for 9% $H_2O_2$ production (Figure S2), with a maximum yield of 0.07 ppb h$^{-1}$, primarily from $O_3$+OLI reactions. This mechanism was found to be significant during winter pollution due to high alkenes and NO concentrations inhibiting $HO_2$ recombination (Qin et al., 2018). Heterogeneous uptake dominated $H_2O_2$ removal, accounting for 69% with a maximum removal rate of 0.7 ppb h$^{-1}$, underscoring its importance during summer pollution periods. Dry deposition, photolysis, and reaction with OH radicals contributed to 25%, 2%, and 4% $H_2O_2$ loss, respectively. These findings provide a comprehensive understanding of $H_2O_2$ sources and sinks in rural environments, emphasizing the critical role of particle uptake in $H_2O_2$ budget.

 **3.5 Precursors control to mitigate H$_2$O$_2$ pollution**

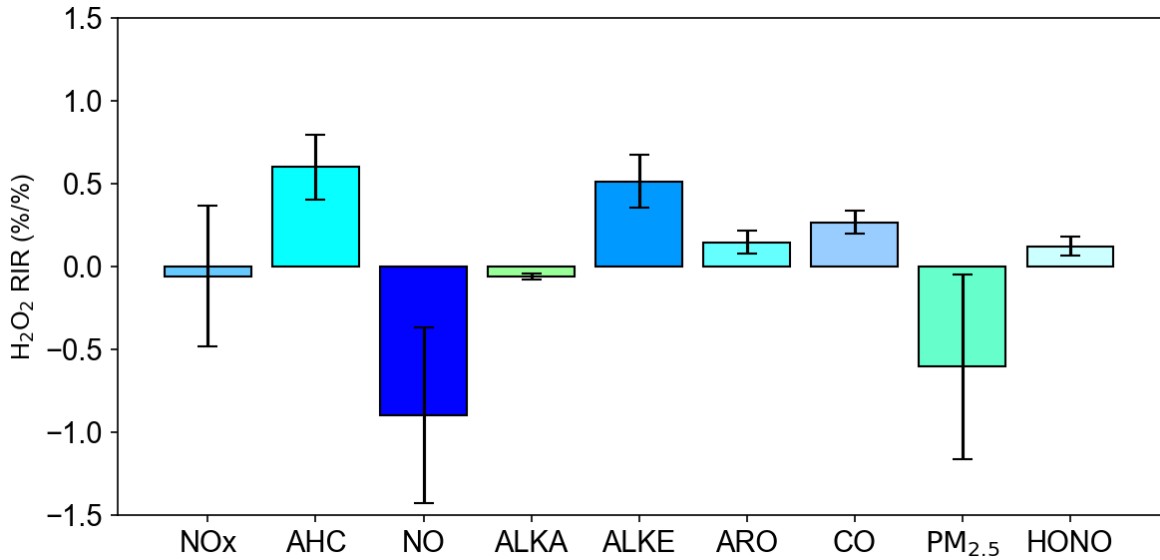

Figure 8. Sensitivity of H$_2$O$_2$ production to different chemical species.

It is evident that photochemical pollution in rural areas is associated with elevated concentrations of H$_2$O$_2$, necessitating
urgent measures to mitigate H$_2$O$_2$ pollution by regulating its precursor compounds. Given the diversity of precursors
involved in H$_2$O$_2$ formation, a critical objective is to quantify the relative contribution of each precursor to H$_2$O$_2$ pollution to
establish prioritized control strategies. In this study, the RIR method was employed to identify the most effective pollutants
for H$_2$O$_2$ control (Figure 8). Here it should be noted that the RIR analysis was performed using the adjusted model with H$_2$O$_2$
uptake coefficient of $6\times10^{-4}$ that showed good agreement with observations. The results demonstrate that reducing NO
concentrations leads to an increase in H$_2$O$_2$ levels, as the reaction between NO and HO$_2$ inhibits H$_2$O$_2$ production. However,
under realistic conditions, a decrease in NO also results in reduced NO$_2$ levels. Since the NO$_2$ heterogeneous reaction is a
significant source of HONO, which serves as a key precursor for OH influencing H$_2$O$_2$ formation, a decline in NO$_2$
consequently reduces H$_2$O$_2$ concentrations. To validate this hypothesis, RIR values for NOx were calculated. Although the
absolute RIR values for NOx remained negative (-0.06), they were significantly lower than those for NO (-0.9), indicating
that the reduction in H$_2$O$_2$ due to decreased NO$_2$ partially offsets the increase in H$_2$O$_2$ caused by reduced NO.

Furthermore, the negative RIR value for alkanes (-0.06) suggests that lowering alkane concentrations enhances H$_2$O$_2$
production, likely due to their lower photochemical reactivities with OH. When alkane levels are reduced, OH radicals
preferentially react with more reactive alkenes and aromatics, leading to increased HO$_2$ and hence more H$_2$O$_2$ formation. The
RIR values for alkenes (0.51), aromatics (0.15), and CO (0.26) were consistently positive, indicating that reducing these
pollutants is effective in reducing H$_2$O$_2$ concentrations, with alkenes exhibiting the most pronounced effect. Consequently,

controlling alkenes concentrations within anthropogenic VOCs should be prioritized, aligning with findings from previous studies (Wang et al., 2016; Ye et al., 2021a). Coal combustion and gasoline exhaust were identified as primary sources of alkenes in the region, underscoring the importance of regulating these emissions to mitigate $H_2O_2$ pollution. Additionally,

RIR value for HONO was 0.12, indicating reducing HONO concentrations can further diminish $H_2O_2$ levels by limiting the primary radical source. Elevated HONO concentrations have been observed across various sites in China, contributing over 40% to primary radical production. Thus, reducing HONO emissions represents a potential mitigating strategy for $H_2O_2$. Ye et al. (2022) reported that HONO emissions due to fertilizer use significantly increase $H_2O_2$ levels in rural areas, suggesting that reducing excessive fertilizer use could mitigate $H_2O_2$ pollution. Moreover, $NO_2$ heterogeneous reactions at various

interfaces and nitrate photolysis are additional sources of HONO (Xue et al., 2020; Xue et al., 2022), highlighting the potential to reduce $H_2O_2$ by decreasing $NO_2$ concentrations and subsequently limiting HONO production.

The RIR value for $PM_{2.5}$ (-0.6) was found to be negative, as reducing $PM_{2.5}$ decreases the uptake of $H_2O_2$, thereby increasing its gas-phase concentration. Recent studies have extensively examined the impact of $PM_{2.5}$ reduction on $O_3$ concentrations,

attributing this phenomenon to diminished $HO_2$ radical uptake and enhanced photolysis rates, both of which elevate $O_3$ levels (Wang et al., 2019; Song et al., 2022). These mechanisms similarly contribute to increased $H_2O_2$ concentrations, yet the effect of particulate matter reduction on $H_2O_2$ has been largely overlooked. This study demonstrates that $PM_{2.5}$ reduction also decreases $H_2O_2$ uptake, further exacerbating its gas-phase concentration. This increase in $H_2O_2$ could enhance sulfate formation efficiency and pose greater threats to human health and ecosystems. Given the critical role of $H_2O_2$ in atmospheric

oxidation capacity, global sulfate aerosol formation, and human health, further research is warranted to investigate $H_2O_2$ trends, environmental impacts, and mitigation strategies.

### 3.6 Implications on $O_3$ formation

$H_2O_2$ measurements serve as a valuable indicator of $O_3$ production sensitivity. Under NOx poor conditions, the $HO_2$ recombination to form $H_2O_2$ represents the primary radical termination pathway. Conversely, under NOx sufficient

conditions, the reaction between $NO_2$ and OH to form nitric acid ($HNO_3$) constitutes the dominant termination mechanism. Sillman (1995) identified the $H_2O_2/HNO_3$ ratio as a robust indicator of $O_3$ sensitivity, with model simulations revealing that a ratio between 0.2 and 0.3 corresponds to a transitional regime, while values exceeding 0.3 indicate NOx-limited conditions and values below 0.2 suggest VOC-limited conditions. In the absence of direct gaseous $HNO_3$ measurements, alternative metrics such as $H_2O_2/NOy$ or $H_2O_2/NOz$ can be employed to assess $O_3$ sensitivity (Sillman et al., 1998), where NOz

encompasses $HNO_3$, PAN, HONO, and alkyl nitrates, and NOy is defined as NOz +NOx.

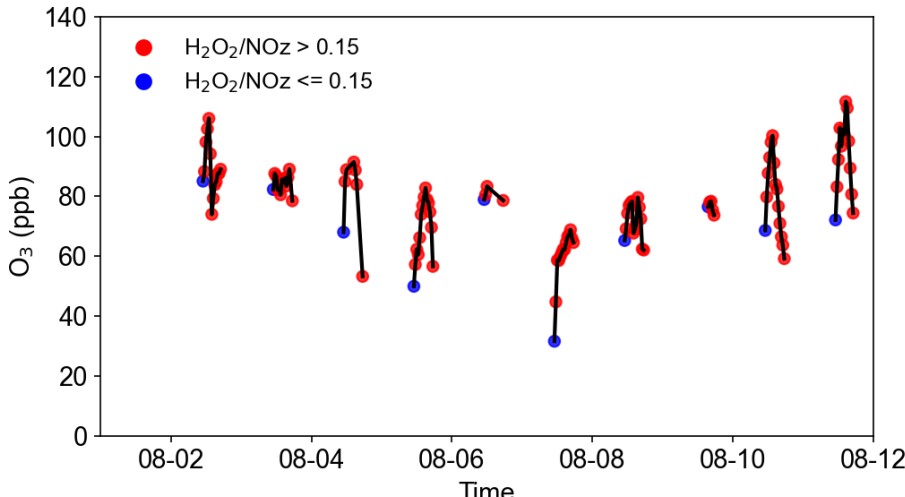

Figure 9. $O_3$ concentrations values from 1 August to 11 August. The red points represent measurements where $H_2O_2/NOz$ is greater than 0.15, while the blue points correspond to measurements where $H_2O_2/NOz$ is less than or equal to 0.15.

In this study, simultaneous measurements of $H_2O_2$ and NOz enabled the determination of $O_3$ sensitivity using the $H_2O_2/NOz$ ratio, with a transitional range identified at 0.15–0.20 (Sillman et al., 1998). The analysis focused on the period of intense photochemical activity between 10:00 and 17:00. As illustrated in Figure 9, over 82% of measured $H_2O_2/NOz$ values exceeded 0.15, indicating that the rural study area predominantly exhibited NOx-limited or transitional conditions during most of the observed period. It is important to note that this metric can be influenced by additional factors. For instance,

significant uptake of $H_2O_2$ by particles was observed in this study, suggesting that the actual photochemical production of $H_2O_2$ is higher than the measured concentrations. Consequently, the theoretical $H_2O_2/NOz$ ratio is likely greater than the observed values, implying that $O_3$ production is more strongly aligned with NOx-limited or transitional regimes.

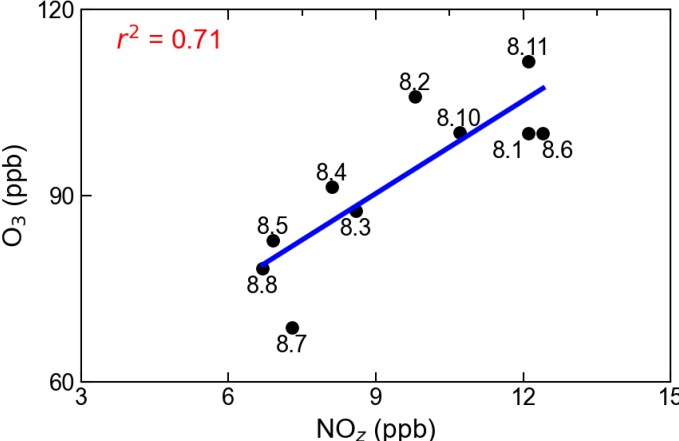

Figure 10. Correlation between daily maxima of $O_3$ and NOz. The numbers adjacent to the solid dots represent the dates.

To corroborate these findings, the $O_3$/NOz ratio was also utilized to evaluate $O_3$ sensitivity. The relationship between peak $O_3$ concentrations and peak NOz concentrations demonstrated a good positive correlation ($r^2$=0.71), with a regression slope of 4.98. This slope is comparable with the value (3.3-7.6) reported in a mountainous area north of Beijing (Wang et al., 2006), but lower than those (6-11) observed in Houston (Daum et al., 2004). Notably, the positive correlation persisted up to NOz concentrations of 12 ppb, differing from observations at other sites where the slope typically decreased for NOz levels

above 10 ppb (Trainer et al., 1993). This deviation can be attributed to reduced $O_3$ production efficiency under VOC-limited conditions. However, the sustained positive correlation across the entire study period suggests that the generation of NOz is consistently accompanied by $O_3$ production, further supporting the prevalence of NOx-sensitive or transitional regimes. These results align with those derived from the $H_2O_2$/NOz ratio, affirming the utility of $H_2O_2$/NOz as a reliable indicator of $O_3$ sensitivity.


The findings underscore the importance of controlling NOx concentrations to mitigate photochemical pollution in rural areas. Tan et al. similarly reported that $O_3$ production in the rural North China Plain is primarily NOx-limited. As NOx emissions continue to decline due to regulatory efforts, an increasing number of regions may transition into NOx-limited or transitional regimes, highlighting the potential benefits of stringent NOx reduction strategies for future $O_3$ pollution control. However,

given the need for synergistic management of $H_2O_2$ and $O_3$, a dual approach targeting both NOx and VOC emissions remains essential. This integrated strategy will be critical for achieving effective and sustainable air quality improvements.

## 4 Conclusions

To investigate photochemical pollution in rural areas, measurements of $H_2O_2$ and related parameters were conducted in the Wangdu region during the summer of 2016. $H_2O_2$ exhibited a distinct diurnal pattern, with an average concentration of

0.62±0.80 ppb. Daily maximum concentrations of $H_2O_2$ varied significantly, ranging from a minimum of 0.2 ppb to a maximum of 4 ppb. The diurnal cycles of $H_2O_2$, PAN, and $O_3$ all followed solar radiation trends, indicating that photochemical reactions predominantly control their production. A good correlation ($r^2$ = 0.55) was observed between daily maximum concentrations of PAN and $O_3$, whereas the correlation between maximum concentrations of $H_2O_2$ and $O_3$ was weak, suggesting that unidentified processes influencing gas-phase $H_2O_2$ concentrations may attenuate this relationship.

Analysis of the $O_3$/$H_2O_2$ ratio revealed that this ratio was significantly higher on polluted days compared to clean days, implying that particle uptake likely reduces gas-phase $H_2O_2$ concentrations.

To further elucidate the factors influencing $H_2O_2$ concentrations, a box model was employed. The model simulations initially overestimated $H_2O_2$ concentrations with a modelled-to-observed ratio of 2.7. However, when $H_2O_2$ heterogeneous uptake

mechanism was incorporated into the model scheme with an uptake coefficient of $6 \times 10^{-4}$, the simulated $H_2O_2$ concentrations aligned well with observed data, underscoring the significant role of heterogeneous uptake in $H_2O_2$ removal. The primary

source of $H_2O_2$ was identified as the bimolecular recombination of $HO_2$, contributing 91% of the total source strength, with a maximum production rate of 1 ppb h$^{-1}$. The dominant removal pathways for $H_2O_2$ included particle uptake (69%), followed by dry deposition (25%), reaction with OH (4%), and photolysis (2%).


Relative Incremental Reactivity (RIR) analysis demonstrated that reducing NOx, PM$_{2.5}$, and alkanes exacerbated $H_2O_2$ concentrations, whereas lowering alkenes, aromatics, CO, and HONO effectively reduced $H_2O_2$ pollution, with alkenes exhibiting the most pronounced impact. The $H_2O_2$/NOz ratio and the positive correlation between daily peak $O_3$ and NOz concentrations indicated that $O_3$ production predominantly occurred in transitional and NOx-limited regimes. To
concurrently mitigate $H_2O_2$ and $O_3$ pollution, a dual strategy focusing on VOC control and stringent NOx reduction is essential. This approach will be critical for achieving synergistic control of photochemical pollutants in rural areas.

Future research should focus on long-term $H_2O_2$ monitoring across different environments in the region, refining the parameterization of heterogeneous uptake processes (particularly for $HO_2$ and $H_2O_2$ under varying aerosol compositions),
and investigating the impacts of changing VOC/NOx ratios on $H_2O_2$ chemistry. In addition, further research on the interactions between gas-phase oxidants and aerosol processes will be vital for understanding the complex feedback mechanisms that influence air quality in rural and urban environments.

**Data availability.** The data used in this study are available from the corresponding author upon request (yjmu@rcees.ac.cn).
**Author contributions.** YM designed the experiments. CY performed $H_2O_2$ measurements and analyzed the data. CY wrote the manuscript with input from PL and CX. All authors contributed to measurements, discussing results, and commenting on the manuscript.
**Competing interests.** The contact author has declared that neither they nor their co-authors have any competing interests.
**Acknowledgements.** We thank the science teams of the summer campaign for their support.
**Financial support.** This work was supported by the National Natural Science Foundation of China (grant nos. 42305099, 42275111).

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
