# Peer review of "Understanding summertime H2O2 chemistry in North China Plain through observations and modelling studies"

_EGUsphere, 2025_

## Author Comment (AC1)

We thank the reviewer for their careful reading of our manuscript and for providing these constructive, specific comments. Below, we respond to each comment point-by-point. Comments by the reviewer are given in black normal font, and our response to the comments is shown in blue. Newly added and modified text in the revised manuscript and supporting information (SI) is given in italics.

**Comment**: Section 3.3: The analysis of $O_3/H_2O_2$ ratio differences between clean and polluted days provides an interesting preliminary assessment, but the authors should acknowledge potential confounding factors like VOC/NOx ratios that might influence this relationship.

**Response**: Thank you for this insightful comment. We agree that the analysis of the $O_3/H_2O_2$ ratio difference between clean and polluted days is a preliminary assessment and that other factors, such as variations in VOC/NOx ratios, could indeed influence this relationship alongside particle uptake effects. We will revise the text in Section 3.3 to explicitly acknowledge these potential confounding factors and state that while the observed difference supports the hypothesis of particle uptake, variations in photochemical regimes (influenced by VOC/NOx) could also contribute.

We have added the following texts:
*Line 300-304: "In addition, differences, potentially driven by varying VOC/NOx ratios between clean and polluted days, could also influence the $O_3/H_2O_2$ relationship independently of particle uptake effects."*

**Comment**: While the study effectively demonstrates the need for particle uptake, the sensitivity to the $HO_2$ uptake coefficient (discussed in lines 304-309) is important. The manuscript correctly notes that this influences the derived $H_2O_2$ uptake coefficient. Perhaps briefly reiterate this uncertainty in the conclusion when stating the $H_2O_2$ uptake coefficient.

**Response**: We agree that reiterating the potential influence of the $HO_2$ uptake uncertainty on the derived $H_2O_2$ uptake coefficient in the conclusion is valuable. We will add a sentence to the abstract to briefly restate this caveat.

We have added the following texts:
*Line 18-20: "A box model with default gas-phase chemistry overestimated $H_2O_2$ by a factor of 2.7, and including particle uptake of $H_2O_2$ (uptake coefficient: $6\times10^{-4}$) improved agreement with observations, although we note this value carries some uncertainty related to the assumed $HO_2$ uptake coefficient."*

**Comment**: The text acknowledges the underestimation of organic peroxides (lines 125-126, 244-245). While $H_2O_2$ is shown to dominate, briefly stating why the method underestimates organic peroxides (e.g., lower collection efficiency for some species) could add clarity for readers unfamiliar with the technique.

**Response**: Thank you for pointing out the need for clarification. We have expanded our explanation of why the measurement method underestimates organic peroxides in Section 2.2:

*Line 126-133: "However, it is important to note that the percentage of organic peroxides reported in this study represents a lower limit, as the collection efficiency of the stripping coil technique varies significantly among different organic peroxide species. While $H_2O_2$ is efficiently trapped due to its high solubility (Henry's law constant: $\sim 10^5$ M atm$^{-1}$), many organic peroxides such as methyl hydroperoxide (MHP) have substantially lower solubilities (Henry's law constant: $\sim 3 \times 10^2$ M atm$^{-1}$), resulting in lower collection efficiencies. Additionally, the catalase used to differentiate between $H_2O_2$ and organic peroxides may not completely discriminate between certain hydroperoxide species, further contributing to uncertainty in organic peroxide quantification."*

**Comment**: There seems to be a minor discrepancy in the stated contribution of particle uptake to $H_2O_2$ loss (69% in Abstract/Conclusions vs. 64% implied by Fig S2 caption/text line 320). Please ensure consistency.

**Response**: We thank the reviewer for catching this inconsistency. We have verified our calculations and the contribution of particle uptake is indeed 69%. We will correct the value mentioned around line 320 (currently 64%) and ensure consistency in the Abstract and Conclusion.

**Comment**: Line 192-196: The manuscript highlights an increasing trend in H2O2 concentrations over time in the North China Plain. While comparisons with previous studies are provided, the discussion on potential drivers of this trend (e.g., changes in NOx/VOC ratios due to emission policies) is limited. The authors should expand on this.

**Response**: This is a very relevant point. We revised the text in Section 3.1 to include a brief discussion on potential drivers.

*Line 201-205: "The significant reduction in NOx emissions in the North China Plain over recent years, while VOC levels remained relatively high or decreased less sharply, has likely shifted the atmospheric chemistry towards conditions more favorable for $HO_2$ recombination, potentially contributing to the observed increasing trend in $H_2O_2$ concentrations. This aligns with the known sensitivity of $H_2O_2$ formation to NOx levels."*

**Comment**: Line 407: The conclusions summarize the key findings well but could include a forward-looking statement on future research needs (e.g., long-term $H_2O_2$ monitoring, improved HO2 uptake parameterization) to guide subsequent studies.

**Response**: Thank you for the suggestion. We added some text at the end of the Conclusion outlining future research needs:

*Line 457-462: "Future research should focus on long-term $H_2O_2$ monitoring across different environments in the region, refining the parameterization of heterogeneous uptake processes (particularly for $HO_2$ and $H_2O_2$ under varying aerosol compositions), and investigating the impacts of changing VOC/NOx ratios on $H_2O_2$ chemistry. In addition, further research on the interactions between gas-phase oxidants and aerosol processes will be vital for understanding the complex feedback mechanisms that influence air quality in rural and urban environments."*

---

## Author Comment (AC2)

We thank Dr. Matthew Johnson for his positive comments and valuable suggestions. We believe addressing these comments will help us to refine our interpretation of the results and identify areas for further clarification. Below, we respond to each comment point-by-point. Comments by the reviewer are given in black normal font, and our response to the comments is shown in blue. Newly added and modified text in the revised manuscript and supporting information (SI) is given in italics.

**Comment**: The authors mention unidentified processes that weaken the $H_2O_2$-$O_3$ correlation. Could they comment on plausible candidates for these processes, such as aqueous-phase reactions or nighttime chemistry? Exploring these possibilities would help clarify what additional mechanisms may need to be included in future modeling efforts.

**Response**: Thank you for your insightful comment regarding the unidentified processes that weaken the $H_2O_2$-$O_3$ correlation, specifically concerning the correlation between their daytime maximum peak concentrations ($r^2 = 0.19$, as shown in Figure 4). We agree with the reviewer that aqueous-phase reactions could also contribute to the weakened $H_2O_2$-$O_3$ correlation. $H_2O_2$ is highly soluble and can undergo various reactions in the aqueous phase (e.g., with dissolved S(IV) or through Fenton-like reactions if transition metals are present), or simply be physically partitioned into the liquid phase. $O_3$, being much less soluble, would be less affected. Day-to-day variations in cloud cover, fog events, or aerosol liquid water content could thus differentially impact the peak concentrations of $H_2O_2$ more significantly than $O_3$, leading to a weaker correlation between their daily maxima. While our current model incorporates a parameterized heterogeneous uptake for $H_2O_2$, it does not explicitly resolve detailed multiphase aqueous chemistry, which could introduce such variability.

The concentrations of $H_2O_2$ and $O_3$ carried over from the previous night set the baseline for the next day's photochemical production. If nighttime loss processes affect $H_2O_2$ and $O_3$ to different extents on different nights (e.g., due to varying NO emissions or aerosol loading/composition), this could alter the net accumulation leading to their respective daily peaks, thereby influencing the day-to-day correlation of these peak values.

In summary, while heterogeneous uptake on particles is a primary candidate we identified, the interplay of aqueous-phase processes, the nuances of nighttime chemistry impacting daytime starting conditions can all contribute to the observed weaker correlation between $H_2O_2$ and $O_3$ daily maximum concentrations.

To address this, we have revised Section 3.3 of the manuscript by adding the following text:

*Line 284-287: "Additionally, aqueous-phase reactions in aerosol water or cloud droplets, facilitated by high relative humidity during the campaign, could further reduce gas-phase $H_2O_2$ without affecting $O_3$, contributing to the decoupling of their peak values. While the focus on daytime maxima limits the direct relevance of nighttime*

*chemistry, processes such as alkene ozonolysis or nocturnal deposition could influence background $H_2O_2$ levels, indirectly affecting daytime peaks."*

**Comment**: The model initially overestimates $H_2O_2$ by a factor of 2.7. How robust are the RIR conclusions in light of this discrepancy? It would be helpful to discuss whether this modeling bias could influence the inferred sensitivity of $H_2O_2$ to different precursors. How sensitive are the model results to the assumed uptake coefficient ($6\times10^{-4}$) for $H_2O_2$? Is there a justification or uncertainty range?

**Response**: We would like to emphasize that the heterogeneous uptake process with a coefficient of $6\times10^{-4}$ was already incorporated into the model when conducting the RIR analysis. The factor of 2.7 overestimation refers to our initial model runs before this adjustment was made. To clarify the sequence of our modeling approach: We first ran the original model which overestimated $H_2O_2$ by a factor of 2.7; we then incorporated heterogeneous uptake with a coefficient of $6\times10^{-4}$ to correct this discrepancy; the RIR analysis was performed using this adjusted model, which showed good agreement with observations. Therefore, the RIR results presented in our manuscript are derived from the optimized model that adequately reproduces the observed $H_2O_2$ concentrations. This means that the initial overestimation did not affect our RIR conclusions.

The following texts was added to avoid confusion:
*Line 358-359: "Here it should be noted that the RIR analysis was performed using this adjusted model with $H_2O_2$ uptake coefficient of $6\times10^{-4}$ that showed good agreement with observations."*

The value of $6\times10^{-4}$ was selected as it leads to the best agreement between observed and simulated $H_2O_2$ concentrations. It falls within the wide range of values reported in laboratory and field studies (typically $10^{-5}$ to $10^{-3}$), which are known to be highly variable depending on factors like aerosol composition, phase, pH, and relative humidity-parameters not fully constrained in our regional simulation. We acknowledge this value carries significant uncertainty. The modeled $H_2O_2$ concentration is sensitive to this parameter. A sensitivity test (Figure S1) indicated that increasing $\gamma$ towards values like $1\times10^{-3}$ could make the modeled $H_2O_2$ concentration 19% lower than the observed $H_2O_2$ concentration. Given the uncertainty and sensitivity, fixing $\gamma$ allows us to explore other aspects of the chemistry, but its accurate representation remains a key challenge. We have clarified the justification and acknowledged the uncertainty and model sensitivity associated with this parameter in Section 3.4.

*Line 319-321: "We believe this value represents a reasonable estimate for the conditions at our sampling site, though we acknowledge that a more dynamic treatment of heterogeneous processes that accounts for variations in aerosol composition, phase state, and ambient RH would be valuable in future studies."*

**Comment**: Does the rural Wangdu site reflect conditions across the North China Plain? How generalizable are the results?

**Response**: Thank you for raising this important question regarding the representativeness of the rural Wangdu site and the generalizability of our results across the North China Plain (NCP). We acknowledge that our study is based on observations from a single rural site in Wangdu, Hebei Province, which may not fully capture the heterogeneity of conditions across the entire NCP. However, we believe that the findings from this site are relevant to a broader context within the NCP for the following reasons, while also recognizing the need for further research to confirm their applicability.

First, the Wangdu site, located in Dongbaituo Village, is surrounded primarily by farmland with no significant nearby industrial facilities, making it representative of typical rural environments in the NCP. The NCP is characterized by extensive agricultural areas interspersed with small villages, and the Wangdu site shares similar land use patterns and emission profiles (e.g., lower NOx compared to urban areas, and contributions from agricultural activities and biomass burning) with many rural areas in the region. It has served as a key location for numerous large-scale campaigns studying regional air pollution in the NCP (e.g., Tan et al., 2017; Peng et al., 2021), suggesting its value for understanding conditions beyond the immediate vicinity.

Second, our results, such as the observed $H_2O_2$ concentrations (average of 0.62 ppb), diurnal patterns aligned with photochemical production, and the predominance of NOx-limited or transitional regimes for $O_3$ formation, are consistent with findings from other rural and suburban sites in the NCP. For instance, Wang et al. (2016) reported comparable $H_2O_2$ levels at the same site in 2014, and studies at other NCP locations, such as Mount Tai (Ye et al., 2021), show similar photochemical behaviors under low NOx conditions. Additionally, the increasing trend of photochemical pollution in rural NCP areas, as noted in our introduction (e.g., Ma et al., 2016), suggests that the processes observed at Wangdu are likely relevant to other rural areas experiencing similar shifts in pollution dynamics due to regional emission reduction policies (e.g., declining NOx and persistent VOC levels).

While the Wangdu site provides a representative case for typical rural environments in the North China Plain, spatial variability in emission sources and atmospheric conditions across the region suggests that further multi-site studies are needed to fully generalize these findings.

**Comment**: In Figure 3, around 19:00, $H_2O_2$ accounts for over 90% of total peroxides, while at 5:00, it accounts for only about 25%. Could the authors comment on the causes of this diurnal variation and the differing behavior of organic peroxides versus $H_2O_2$? This would enrich the interpretation of the peroxide measurements and their photochemical dynamics.

**Response**: At 19:00, following peak daytime photochemical activity, the highly

efficient $HO_2+HO_2$ pathway has led to substantial $H_2O_2$ accumulation. While organic peroxides (ROOH) have also been produced, the sheer rate of the $HO_2$ self-reaction often makes $H_2O_2$ the dominant peroxide species generated during intense daytime photochemistry. At this time, photochemical production is ceasing, but the accumulated $H_2O_2$ constitutes a large fraction (>90%) of the total peroxide.

Overnight, photochemical production has stopped entirely. Both $H_2O_2$ and ROOH concentrations decrease due to deposition and potential heterogeneous losses. In contrast, some longer-lived ROOH species formed during the previous day might persist relatively well for low dry deposition velocities. For instance, the dry deposition velocity of $CH_3OOH$ is 30 times smaller than that of $H_2O_2$. This combination of continuous $H_2O_2$ loss and the persistence ROOH species leads to a relative increase in the contribution of ROOH to the (lower) total peroxide concentration observed in the early morning (~25% H2O2 vs ~75% ROOH in our case).

To address this comment, we have added the following text to the revised manuscript:

*Line 259-267: "The diurnal variation in the relative contributions of $H_2O_2$ and organic peroxides to total peroxides, reflects their distinct production and loss mechanisms. $H_2O_2$ dominates (over 90%) around 19:00 due to strong photochemical production via $HO_2$ recombination during the day, while its contribution drops to ~25% by 05:00 due to nighttime losses (e.g., heterogeneous uptake and dry deposition) without replenishment. In contrast, organic peroxides contribute more significantly in the early morning, likely due to slower loss rates compared to $H_2O_2$. Organic peroxides such as $CH_3OOH$ (methyl hydroperoxide) have much lower dry deposition rates— approximately 30 times lower than that of $H_2O_2$-leading to less nighttime loss and a higher relative contribution to total peroxides during early morning hours. These differences highlight the distinct photochemical dynamics and loss mechanisms of $H_2O_2$ compared to organic peroxides, influenced by diurnal variations in radiation, precursor concentrations, and meteorological conditions."*

**Comment**: Line 151: Please change lagrangian to Lagrangian, as it is a proper adjective derived from Joseph-Louis Lagrange (analogous to Watt, Poisson, Newtonian, etc.).

**Response**: We have changed accordingly.

**Comment**: Typographic conventions: According to the IUPAC Green Book (3rd edition, 2007), symbols for physical quantities should be printed in italic type to distinguish them from unit symbols. Please revise:

k in line 158
$r^2$ in lines 258, 259, and Figure 10
T, P, and other physical quantities, if applicable elsewhere
Line 174 and elsewhere: Change O1D to $O(^1D)$ to reflect the correct notation for electronically excited oxygen.

Line 177: Use a subscript for O3-i.e., O3.

Line 189: Change Hongkong to Hong Kong, the correct spelling in English.

Line 230: Change O3P to O(3P) to properly denote the electronic state of ground-state atomic oxygen.

Response: Thank you for pointing out these mistakes. All corrected as suggested.

References:

Wang, Y., Chen, Z. M., Wu, Q. Q., Liang, H., Huang, L. B., Li, H., Lu, K. D., Wu, Y. S., Dong, H. B., Zeng, L. M., and Zhang, Y. H.: Observation of atmospheric peroxides during Wangdu Campaign 2014 at a rural site in the North China Plain, Atmos Chem Phys, 16, 10985-11000, 10.5194/acp-16-10985-2016, 2016.

Ye, C., Xue, C., Zhang, C., Ma, Z., Liu, P., Zhang, Y., Liu, C., Zhao, X., Zhang, W., He, X., Song, Y., Liu, J., Wang, W., Sui, B., Cui, R., Yang, X., Mei, R., Chen, J., and Mu, Y.: Atmospheric Hydrogen Peroxide (H2O2) at the Foot and Summit of Mt. Tai: Variations, Sources and Sinks, and Implications for Ozone Formation Chemistry, Journal of Geophysical Research: Atmospheres, 126, e2020JD033975, https://doi.org/10.1029/2020JD033975, 2021a.

Ma, Z., Xu, J., Quan, W., Zhang, Z., Lin, W., and Xu, X.: Significant increase of surface ozone at a rural site, north of eastern China, Atmos. Chem. Phys., 16, 3969-3977, 10.5194/acp-16-3969-2016, 2016.

---

## Author Comment (AC3)

[revised manuscript text omitted]
υ→OH+NO                                                                                            R0

$$VOCs+OH\rightarrow RO_2+H_2O \qquad\qquad\qquad R1$$
$$RO_2+NO\rightarrow HO_2+RO_2+OVOC \qquad\qquad\qquad R2$$
$$HO_2+NO\rightarrow NO_2+OH \qquad\qquad\qquad R3$$
$$NO_2+h\upsilon\rightarrow NO+O(^3P) \qquad\qquad\qquad R4$$
$$O(^3P)+O_2\rightarrow O_3 \qquad\qquad\qquad R5$$
$$HO_2+HO_2+M\rightarrow H_2O_2+H_2O+M \qquad\qquad\qquad R6$$
$$CH_3C(O)OO+NO_2+M\rightarrow CH_3C(O)OONO_2(PAN)+H_2O+M \qquad\qquad\qquad R7$$

Following their peaks, the concentrations of all three oxidants declined rapidly. For $H_2O_2$, this decrease was primarily driven by dry deposition and, in the evening, enhanced uptake by liquid aerosols formed as relative humidity increased. $O_3$ concentrations dropped due to a combination of dry deposition and NO titration, while PAN levels decreased mainly through thermal decomposition. At night, the absence of photochemical reactions caused all three oxidants to maintain low concentrations.

[Figure]

Figure 3. The concentrations of $H_2O_2$ and organic peroxides.

[revised manuscript text omitted]

---

## Author Response (AR2)

Dear editor,

Thank you for the constructive comments on our manuscript. We have carefully addressed your concerns and made appropriate revisions. Below, comments by the editor are given in black normal font, and our response is shown in blue. Newly added and modified text in the revised manuscript is given in italics.

**Comment**: Figure 2 and 3 captions: "Observed" diurnal cycles "averaged over the whole campaign period"? A more detailed explanation is preferred.

**Response**: We have revised the captions for Figure 2 and Figure 3 to explicitly state that these are average diurnal cycles based on observations collected throughout the entire campaign period.

Line 228:
*"Average diurnal cycles of $H_2O_2$, PAN, and $O_3$ observed throughout the entire campaign period at the SRE-RCEES site."*

Line 248:
*"Average diurnal cycles of $H_2O_2$ and organic peroxides (ROOH) observed throughout the entire campaign period at the SRE-RCEES site."*

**Comment**: The explanation of the $H_2O_2$ processes is almost good, but the reason for the minimum of organic peroxides around 19:00 has not been well explained. Is this due to instrument issues or atmospheric processes?

**Response**: Thank you for highlighting the need for a clearer explanation regarding the minimum concentration of organic peroxides around 19:00. The observed minimum in organic peroxide concentrations around 19:00 is primarily attributed to atmospheric processes rather than instrument issues. The minimum in ROOH concentration observed around 19:00 represents a transitional point. By this time, daytime photochemical production has largely ceased due to diminishing solar radiation, leading to a decline from its afternoon peak as removal processes continue. The subsequent increase in ROOH concentration after 19:00, which makes 19:00 a local minimum, may be attributed to nighttime chemical production primarily through (a) the ozonolysis of alkenes ($O_3$ + alkenes → … → $RO_2$ → ROOH), and (b) $NO_3$ radical-initiated oxidation of VOCs ($NO_2$ + $O_3$ → $NO_3$; $NO_3$ + VOCs → … → $RO_2$ → ROOH). These processes become major sources of $RO_2$ (and subsequently ROOH) during the night. In contrast, $H_2O_2$ typically continues to decrease throughout the night. Although ozonolysis can also be a source $H_2O_2$, $H_2O_2$ generally has a higher dry deposition velocity than many ROOH species, leading to more efficient net removal overnight.

Line 266-274:
*"The minimum in ROOH concentration observed around 19:00 represents a*

*transitional point. By this time, daytime photochemical production has largely ceased due to diminishing solar radiation, leading to a decline from its afternoon peak as removal processes continue. The subsequent increase in ROOH concentration after 19:00, which makes 19:00 a local minimum, may be attributed to nighttime chemical production primarily through (a) the ozonolysis of alkenes ($O_3$ + alkenes → ... → $RO_2$ → ROOH), and (b) $NO_3$ radical-initiated oxidation of VOCs ($NO_2 + O_3 → NO_3$; $NO_3$ + VOCs → ... → $RO_2$ → ROOH). These processes become major sources of $RO_2$ (and subsequently ROOH) during the night. In contrast, $H_2O_2$ typically continues to decrease throughout the night. Although ozonolysis can also be a source $H_2O_2$, $H_2O_2$ generally has a higher dry deposition velocity than many ROOH species, leading to more efficient net removal overnight."*